# Role of Inflammation and Redox Status on Doxorubicin-Induced Cardiotoxicity in Infant and Adult CD-1 Male Mice

**DOI:** 10.3390/biom11111725

**Published:** 2021-11-19

**Authors:** Ana Reis-Mendes, Ana Isabel Padrão, José Alberto Duarte, Salomé Gonçalves-Monteiro, Margarida Duarte-Araújo, Fernando Remião, Félix Carvalho, Emília Sousa, Maria Lourdes Bastos, Vera Marisa Costa

**Affiliations:** 1Associate Laboratory i4HB-Institute for Health and Bioeconomy, Laboratory of Toxicology, Department of Biological Sciences, Faculty of Pharmacy, University of Porto, 4050-313 Porto, Portugal; remiao@ff.up.pt (F.R.); felixdc@ff.up.pt (F.C.); mlbastos@ff.up.pt (M.L.B.); 2UCIBIO-Applied Molecular Biosciences Unit, REQUIMTE, Laboratory of Toxicology, Department of Biological Sciences, Faculty of Pharmacy, University of Porto, 4050-313 Porto, Portugal; 3Research Center in Physical Activity, Health and Leisure (CIAFEL), Laboratory for Integrative and Translational Research in Population Health (ITR), Faculty of Sport, University of Porto, 4200-450 Porto, Portugal; apadrao@fade.up.pt (A.I.P.); jarduarte@fade.up.pt (J.A.D.); 4TOXRUN–Toxicology Research Unit, University Institute of Health Sciences, Advanced Polytechnic and University Cooperative (CESPU), CRL, 4585-116 Gandra, Portugal; 5Outcomes Research Laboratory, MOREHealth, Outcomes Research Laboratory, Portuguese Institute of Oncology at Porto Francisco Gentil (IPO Porto), 4200-072 Porto, Portugal; salomemonteiro8180@gmail.com; 6Department of Immuno-Physiology and Pharmacology, ICBAS—Institute of Biomedical Sciences Abel Salazar, University of Porto, 4050-313 Porto, Portugal; mdcma@icbas.up.pt; 7Laboratory of Organic and Pharmaceutical Chemistry, Chemistry Department, Faculty of Pharmacy, University of Porto, 4050-313 Porto, Portugal; esousa@ff.up.pt; 8CIIMAR–Interdisciplinary Centre of Marine and Environmental Research, 4450-208 Porto, Portugal

**Keywords:** doxorubicin, mice, age, cumulative dose, cardiotoxicity, oxidative stress, inflammation

## Abstract

Doxorubicin (DOX) is a topoisomerase II inhibitor commonly used in the treatment of several types of cancer. Despite its efficacy, DOX can potentially cause fatal adverse effects, like cardiotoxicity. This work aimed to assess the role of inflammation in DOX-treated infant and adult mice and its possible link to underlying cardiotoxicity. Two groups of CD-1 male mice of different ages (infants or adults) were subjected to biweekly DOX administrations, to reach a cumulative dose of 18.0 mg/kg, which corresponds approximately in humans to 100.6 mg/m^2^ for infants and 108.9 mg/m^2^ for adults a clinically relevant dose in humans. The classic plasmatic markers of cardiotoxicity increased, and that damage was confirmed by histopathological findings in both groups, although it was higher in adults. Moreover, in DOX-treated adults, an increase of cardiac fibrosis was observed, which was accompanied by an increase in specific inflammatory parameters, namely, macrophage M1 and nuclear factor kappa B (NF-κB) p65 subunit, with a trend toward increased levels of the tumor necrosis factor receptor 2 (TNFR2). On the other hand, the levels of myeloperoxidase (MPO) and interleukin (IL)-6 significantly decreased in DOX-treated adult animals. In infants, a significant increase in cardiac protein carbonylation and in the levels of nuclear factor erythroid-2 related factor 2 (Nrf2) was observed. In both groups, no differences were found in the levels of tumor necrosis factor (TNF-α), IL-1β, p38 mitogen-activated protein kinase (p38 MAPK) or NF-κB p52 subunit. In conclusion, using a clinically relevant dose of DOX, our study demonstrated that cardiac effects are associated not only with the intensity of the inflammatory response but also with redox response. Adult mice seemed to be more prone to DOX-induced cardiotoxicity by mechanisms related to inflammation, while infant mice seem to be protected from the damage caused by DOX, possibly by activating such antioxidant defenses as Nrf2.

## 1. Introduction

Doxorubicin (DOX) is a topoisomerase II inhibitor that binds and intercalates with the DNA, with a broad spectrum of activity against neoplastic cells [1]. This drug is widely used in chemotherapy regimens for solid tumors and hematological malignancies, including breast, prostate, uterus, ovary, esophagus, liver, bile ducts and stomach tumors, childhood tumors (e.g., Wilms’ tumor), osteosarcomas and soft tissue sarcomas, Kaposi’s sarcoma, multiple myeloma, Hodgkin’s and non-Hodgkin’s lymphoma [2]. Nevertheless, adverse side effects come with its use, cardiotoxicity being a limiting effect with serious and possibly life-threatening consequences. This is often characterized by arrhythmias, electrocardiographic changes (non-specific ST-T changes and QT prolongation) and heart failure (HF) [2]. The usual intravenous (IV) administration schedule of DOX is 60–75 mg/m^2^ every 3 weeks, depending on the type of tumor and patients’ characteristics, among other factors. Nevertheless, the total lifetime cumulative dose of 400–550 mg/m^2^ should not be surpassed, as it exponentially increases the risk of cardiotoxicity [2]. The total cumulative dose is the most powerful predictor for the development of DOX-induced cardiotoxicity. It has been estimated that 3% of patients develop DOX-related HF after a cumulative dose of 400 mg/m^2^, 7% after 550 mg/m^2^, and 18% after 700 mg/m^2^ [3]. Other authors estimated a higher incidence of cardiotoxicity in comparison with the retrospective study by Hon Hoff et al. [3]: 5% of patients after a cumulative dose of 400 mg/m^2^; 26% of patients after 550 mg/m^2^, and 48% of patients after 700 mg/m^2^ [4]. The patients’ age is also a risk factor, as children seem to be at greater risk of developing cardiac toxicity. The estimated risk of HF development in children increases with time, affecting 5.5% of patients 20 years after the beginning of therapy, unless the cumulative dose reaches 300 mg/m^2^ or more, in which case, the risk of HF approaches 10% [5]. However, even children exposed to lower doses are at risk of developing HF [6]. While DOX exposure has been assumed to have similar acute cellular effects on the heart, regardless of age, there is mounting evidence that highlights the significant clinical differences between pediatric and adult DOX-induced cardiotoxicity. Distinct molecular and cellular effects could influence everything from the mechanisms of acute damage, injury repair, and pathologic progression to overt cardiac disease [7].

Over the years, several key events have been suggested as being associated with DOX-induced cardiotoxicity: oxidative stress, the inhibition of nucleic acid and protein synthesis, the release of vasoactive substances (histamine and catecholamines), the altered function of myocardial adrenergic receptors and adenylate cyclase activity, changes in lysosomal morphology and enzyme activities, disturbed calcium transport in cardiac sarcolemma, iron accumulation in mitochondria, the presence of cardiotoxic metabolites, the induction of apoptosis, or the inhibition of cardiac topoisomerase 2β [8]. Nevertheless, despite efforts to unravel the molecular mechanisms responsible for DOX-induced cardiotoxicity, biomarkers or effective therapeutic alternatives have not yet been found.

Inflammation has been implicated in DOX-induced cardiotoxicity [9,10,11,12]. DOX causes inflammatory effects in the myocardium by the activation of nuclear factor kappa B (NF-κB), a critical regulator of inflammatory and immunological reactions [13,14,15]. The relation between inflammation and HF is complex and bidirectional, since HF-related hemodynamic stress induces a state of inflammation that triggers the release of proinflammatory cytokines by cardiomyocytes and cardiac fibroblasts, including tumor necrosis factor (TNF-α), interleukin (IL)-6, IL-1β, angiotensin II, and myostatin, which impair cardiac structure and function [16]. Furthermore, HF triggers mitochondrial dysfunction, generates reactive oxygen species (ROS) and leads to the activation of the NLRP3 inflammasome, with the consequent maturation of proinflammatory cytokines, such as IL-1β and IL-18 [16].

Considering the existing knowledge, this study aimed to evaluate the role of inflammation in DOX-induced cardiotoxicity, at a therapeutically relevant dose, in infant and adult mice.

## 2. Materials and Methods

All chemicals and reagents were of analytical grade or of the highest grade available. Doxorubicin hydrochloride (≥98% purity, DOX) DL-dithiothreitol (DTT), phenylmethane sulfonyl fluoride (PMSF), a protease inhibitors cocktail, sodium fluoride (NaF), sodium metavanadate (NaVO_3_), Ponceau S, Tween20 and 5,5′-dithiobis(2-nitrobenzoic acid) (DTNB), adenosine 5′-triphosphate (ATP), disodium salt hydrate, reduced glutathione (GSH), glutathione reductase, glutathione oxidized disodium salt, 2-vinylpyridine, paraformaldehyde, bovine serum albumin (BSA), H_2_O_2_ 30% (*v*/*v*), trisodium citrate dehydrate, direct red 80, and the PAP pen for immunostaining were obtained from Sigma-Aldrich (St. Louis, MO, USA). β-Nicotinamide adenine dinucleotide 2′-phosphate reduced (NADPH) tetrasodium salt hydrate was obtained from PanReac AppliChem ITW Reagents (Barcelona, Spain). Ethylenediaminetetraacetic acid (EDTA), perchloric acid, sodium hydroxide, magnesium chloride, sodium carbonate, disodium phosphate, copper (II) sulfate, potassium bicarbonate, potassium dihydrogen phosphate, magnesium sulfate, potassium chloride, Histosec paraffin pastilles and FolinCiocalteu reagent were obtained from Merck (Darmstadt, Germany). A Bio-Rad DC protein assay kit was obtained from Bio-Rad Laboratories (Hercules, CA, USA). Phosphate buffered saline solution (PBS) was obtained from Biochrom (Berlin, Germany), sodium chloride (NaCl), and sodium dodecyl sulfate (SDS) from VWR (Leuven, Belgium), potassium sodium tartrate from Fluka (Buchs SG, Switzerland), methanol, DPX mounting media, and xylene from Thermo Fisher Scientific (Loughborough, UK). Harris hematoxylin was obtained from Harris Surgipath (Richmond, IL, USA), 1% aqueous eosin from Australian Biostain (Traralgon, Australia), and isoflurane (Isoflo^®^) was obtained from Abbott Animal Health (North Chicago, IL, USA). ABX Pentra reagents were obtained from HORIBA (Kyoto, Japan). Enhanced chemiluminescence (ECL) reagents, and 0.45 µm Amersham Protran nitrocellulose blotting membrane were obtained from GE Healthcare Bio-Sciences (Pittsburgh, PA, USA). Rabbit polyclonal anti-IL-6 (ab83339), rabbit polyclonal anti-NF-κB p65 (ab16502), mouse monoclonal anti-NF-κB p100/p52 (ab71108), rabbit polyclonal anti-TNF-α (ab66579), rabbit polyclonal anti-myeloperoxidase (ab139748), rabbit polyclonal anti-mannose receptor (M2 macrophage, ab64693), rabbit polyclonal anti-CD68 (M1 macrophage, ab125212), goat anti-rabbit IgG-horseradish peroxidase (ab97051) and goat anti-mouse IgG-horseradish peroxidase (ab6728) from Abcam (Cambridge, UK). Rabbit polyclonal anti-IL-1β (P420B) was obtained from Thermo Fisher Scientific (Waltham, MA, USA) and rabbit polyclonal anti-p38 MAPK (#9212) was purchased from Cell Signaling Technology. Rabbit polyclonal dinitrophenyl (DNP)-KLH (A6430) was obtained from Invitrogen/Life Technologies (Grand Island, NY, USA). A 3,3′-diaminobenzidine (DAB, HIGHDEF^®^ DAB chromogen/substrate set was obtained from Enzo Life Sciences (Miraflores, Portugal). Water was purified with a Milli-Q Plus ultrapure water purification system (Millipore, Bedford, MA, USA).

### 2.1. Animals

Male CD-1 mice (*Mus musculus*) were born at Charles River Laboratories (L’Arbresle, France) and kept at the Institute of Biomedical Sciences Abel Salazar, University of Porto (ICBAS-UP) animal house facility, always accompanied by experienced veterinarians until the day of the sacrifice. Animals were housed in IVC Sealsafe Plus Green Mouse 500 cages, with controlled temperature (22 ± 2 °C) and humidity, in a 12-hour light-dark cycle. Certified rodent diet (4RF21 GLP, Mucedola, Settimo Milanese, Italy) and autoclaved water were provided ad libitum, and each cage had environmental enrichment. Animals were acclimated for seven days before starting the experiments. Animals were housed according to European Union recommendations and the revision of Appendix A of the European Convention ETS 123. The protocol was approved by the local animal welfare body (ICBAS-UP ORBEA, ref. 140/2015) and the Portuguese national authority for animal health (DGAV, ref. 021322 of 26 October 2016).

### 2.2. Experimental Protocol

Mice in the infant group (*n* = 12) weighed 10–12 g and were 3 weeks old, corresponding to approximately 180 human days, according to the literature [17]. Mice in the adult group (*n* = 14) weighed 38–56 g, were 12 weeks old and had clearly reached early adulthood, an age corresponding to approximately 20 human years [17]. The animals were further divided into two subgroups of infant mice (6 infant DOX-treated mice (I-DOX) and 6 control mice (I-Control)) and into two subgroups of adult mice (7 adult DOX-treated mice (A-DOX) and 7 control mice (A-Control)).

### 2.3. DOX Administration Schedule

To mimic human DOX-therapy, the drug was administered in cycles, with administrations being interrupted by drug-free periods (Figure 1). Thus, mice were treated biweekly with an intraperitoneal (IP) injection with saline solution (NaCl 0.9%, control group) or DOX (DOX group) over three weeks, to reach a total dosage of 18.0 mg/kg. This total dose of DOX corresponds approximately in humans to 100.6 mg/m^2^ for infants and 108.9 mg/m^2^ for adults [18,19,20]. These doses did not exceed the maximum dosage of DOX recommended in humans [2] and are similar between groups.

In humans, DOX is administered via IV, which can cause tissue necrosis [21]. In this study, we decided to give DOX via the IP route, thus avoiding local complications and causing less discomfort to the animals [22]. The total dose was then divided into 6 IP administrations (2 per week, for 3 weeks), alternating between the left and right sides of the abdomen. During the experimental period, food and water intake, body weight and animal welfare were assessed daily. The food and water were provided for each cage and the amount consumed was based on the individual animal body weight, since the animals were maintained in a social environment. For sacrifice, infant mice were fully anesthetized with 5% isoflurane and exsanguinated 17 days after the last administration (which correspond for infants to ~1.07 human years) while adults suffered the same procedure 7 days (which correspond for adults to ~1.92 human years) after the last administration. Animals’ well-being was considered through a “scoring system” previously described [18] to assess and minimize their suffering and stress. It is of relevance to mention that, within those periods of sacrifice and after the corresponding correction to “human time” [17], cardiac dysfunction has been described in humans [23].

### 2.4. Blood Collection and Measurement of Plasma Biomarkers

Blood was collected from the inferior vena cava and transferred to EDTA-containing tubes. After centrifugation (920× *g*, 10 min, 4 °C), the obtained plasma was stored at −20 °C until the evaluation of plasmatic biomarkers. The activity of aspartate aminotransferase (AST), alanine aminotransferase (ALT), creatine kinase-MB (CK-MB), and total creatine-kinase (total-CK) was determined through enzymatic assays in the apparatus ABX Pentra 400 with ABX Pentra reagents, according to the manufacturer’s instructions.

### 2.5. Tissue Collection

The heart and brain were removed, weighed, and used to assess the ratio of heart weight to brain weight. The hearts were processed as follows: segments of heart tissue were placed in 4% paraformaldehyde, pH 7.2–7.4, and used for histological and immunohistochemistry analysis. A section of the heart was placed in complete RIPA buffer [50 mM Tris-HCl, 150 mM NaCl, 1% Triton X-100 (*v*/*v*), 0.5% sodium deoxycholate (*w*/*v*), and 0.1% SDS (*w*/*v*), pH 8.0, supplemented with 0.25 mM PMSF, 1 mM NaVO_4_, 10 mM NaF, 1 mM DTT and 0.5% (*v*/*v*) complete protease inhibitor cocktail] and stored at −80 °C for immunoblotting analysis. Total protein was then quantified using the Bio-Rad DC Protein assay. The remaining sections of the heart were homogenized in an ice-cold 0.1 M phosphate-buffered (pH 7.4) solution with an Ultra-Turrax^®^ Homogenizer. The homogenate was aliquoted and acidified to a final concentration of 5% perchloric acid. Following centrifugation, the supernatant obtained was used for the measurements of ATP, total glutathione (tGSH), GSH, and oxidized glutathione (GSSG) levels. All steps were performed in ice. An aliquot of homogenate was used to quantify the protein levels using the Lowry method [24].

### 2.6. Histological Analysis of Cardiac Tissue

Histological procedures were performed as previously reported [18]. Hematoxylin and eosin staining was performed for assessing tissue damage. To examine vascular and interstitial fibrosis, Sirius Red staining was performed. For both staining’s, the slides were examined and photographed with a Carl Zeiss Imager A1 light microscope equipped with an AxioCam MRc 5 digital camera (Oberkochen, Germany). Histopathological evidence of tissue damage was calculated in terms of their severity and incidence in every slide, according to what has been previously published [18,25]. To semi-quantify the severity of the damage caused to the cardiac tissue, the samples were microscopically characterized in a blinded fashion, according to the following parameters: (i) cellular degeneration, (ii) interstitial inflammatory cell infiltration, (iii) necrotic zones, and (iv) loss of tissue organization. The severity of cellular degeneration was scored according to the number of cells that showed alterations (dilatation, vacuolization, pyknotic nuclei, and cellular density) in the light microscopy visual field, using a scale from 0 to 3. Tissue necrosis severity, tissue disorganization and inflammatory activity were also scored (from 0 to 3 values) according to the amount of tissue affected [18,25]. Sections stained with Sirius Red were evaluated using ImageJ software (NIH, Bethesda, MD, USA). The results of collagen seen as red staining are expressed as a percentage of collagen per total section area, as previously detailed [18].

### 2.7. Measurement of ATP Levels

ATP levels were determined by a bioluminescent assay based on the luciferin-luciferase reaction, as previously described [25,26]. The results were expressed as nmol of ATP per mg of protein (nmol ATP/mg protein).

### 2.8. Measurement of tGSH, GSH, and GSSG

GSHt and GSSG levels were evaluated by the DTNB-GSSG reductase recycling assay, as previously described [25,26]. The levels of GSH were calculated using the formula: GSH = GSHt − 2 × GSSG. The results of GSHt, GSH and GSSG were normalized to the total protein content and expressed as nmol of GSH or GSSG per mg of protein (nmol GSH/mg protein or nmol GSSG/mg protein).

### 2.9. Protein Carbonylation Evaluation

Heart sections were lysed in RIPA buffer through sonication and were kept at −80 °C until analysis. Samples containing 20 µg of protein were then processed, as previously described [27]. The slot-blot membranes were blocked with 5% (*w*/*v*) dry non-fat milk in TBS-T (100 mM Tris, 1.5 mM NaCl, pH 8.0 and 0.5% Tween 20) for 1 h and then incubated for 1 h with primary antibody (rabbit anti-DNP, 1:1000) in 5% (*w*/*v*) non-fat dry milk in TBS-T. The membranes were washed three times (10 min each) with TBS-T and incubated for 1 h with a solution of anti-rabbit IgG-peroxidase (1:5000). Immunoreactive bands were detected using the ECL reagents, according to the supplier’s instructions, and digital images were acquired using the ChemiDoc Imaging System version 2.3.0.07 (Bio-Rad, Hercules, CA, USA). The images obtained were analyzed using Image Lab software version 6.0.1 (Bio-Rad, Hercules, CA, USA).

### 2.10. Immunohistochemistry

The detection of NF-κB p65, M1 and M2 macrophages in the heart was performed as previously described [18]. All preparations were analyzed in a Carl Zeiss Imager A1 light microscope and images were recorded using a coupled AxioCam MRc5 digital camera (Oberkochen, Germany).

### 2.11. Western Blotting Analysis

Western blot analysis was performed as previously reported [18]. Heart tissue was lysed using the RIPA lysis buffer, supplemented with PMSF, DTT, NaF, NaVO_3_ and a cocktail of protease inhibitors, as mentioned. The supernatant was collected, and the total protein was quantified using the Bio-Rad DC Protein assay. Forty µg of protein were loaded into electrophoresis on a 12.5% SDS-PAGE gel. Gels were blotted onto a nitrocellulose membrane (Amersham Protran, GE Healthcare, Germany) in a transfer buffer (25 mM Tris, 192 mM glycine, pH 8.3 and 20% methanol) for 2 h (200 mA). Then, nonspecific binding was blocked with 5% (*w*/*v*) dry non-fat milk in TBS-T (100 mM Tris, 1.5 mM NaCl, pH 8.0 and 0.5% Tween 20). Membranes were incubated with primary antibodies diluted to 1:1000 (rabbit anti-p38 MAPK, rabbit anti-NF-κB p65, mouse anti-NF-κB p52, rabbit anti-IL-1 beta) or diluted to 1:500 (rabbit anti-TNF-α, rabbit anti-IL-6, rabbit anti-myeloperoxidase) in 5% (*w*/*v*) non-fat dry milk in TBS-T for 2 h at room temperature or overnight. Then, the membranes were washed and incubated with secondary horseradish peroxidase-conjugated anti-rabbit (1:10,000) or anti-mouse (1:5000). Immunoreactive bands were detected by using the enhanced chemiluminescence ECL (Amersham Pharmacia Biotech, Buckinghamshire, UK) according to the manufacturer’s instructions. The immunoreactive bands were automatically detected using the ChemiDoc Imaging System version 2.3.0.07 (Bio-Rad, Hercules, CA, USA). The images obtained were analyzed using the Image Lab software version 6.0.1 (Bio-Rad, Hercules, CA, USA). Protein loading was confirmed by Ponceau S staining.

### 2.12. Statistical Analysis

Results are expressed as mean ± standard deviation (SD). Statistical analyses of the animal weight, food, and water intake data were carried out by the two-way analysis of variance (two-way ANOVA) followed by the Sidak *post hoc* test. The outliers were identified using the ROUT method (Q = 1%). When two groups and one time point were analyzed, statistical analysis was performed by Student’s *t*-test when the distribution was normal or by a Mann–Whitney test when the distribution was not normal. Statistical significance was considered to be *p* < 0.05. To perform the statistical analysis, the GraphPad Prism 8.0 software program (San Diego, CA, USA) was used.

## 3. Results

### 3.1. DOX Treatment Affected Mice Body Weight, Food/Water Consumption, and the Heart Weight-to-Brain Weight Ratio

In the infant group, DOX-treated mice had significantly less body weight gain compared to controls from the 17th day (Figure 2A) onwards. Concerning adult mice, there was a significant bodyweight decrease in the DOX-treated animals compared to controls from day 17 onwards (Figure 2B). Both DOX-treated groups showed a progressive decrease in body condition, had piloerection, weakness, and a hunched posture. To avoid prolonging animals’ discomfort, sacrifice was performed when critical scores were achieved (through a “scoring system” previously described [18]): one week for adults and 17 days in infants. In the DOX-treated adult mice group, the necropsy performed after euthanasia revealed, in some animals, the presence of ascites, a dilated stomach and reduced liver weight.

As it is impossible to know the exact food and water consumption of each animal and at the same time maintain a social environment. For this reason, all consumptions were normalized to their weights, assuming that animals had intake rates proportional to their current weights. Concerning food consumption, a significant decrease in food intake in the infant (Figure 2C) and adult (Figure 2D) groups treated with DOX from the 10th day until the end of the experiment was observed, compared to the control groups.

In the infant group, DOX-treated mice showed a significant decrease in water consumption from day 14 onwards, whereas, in adults, the water intake decreased significantly in the last days of the protocol, compared to controls (Figure 2F).

As shown in Table 1, our data showed a significant (*p* < 0.01) decrease in the heart weight-to-brain weight ratio in DOX-treated mice, both in infants and adults, when compared to controls.

### 3.2. Plasma Aspartate Aminotransferase Was Found to Be Increased in Both Groups, While Creatine Kinase-MB and Total Creatine-Kinase Only Increased in Adult Mice

The main serum markers of myocardial damage were examined, and the results are shown in Table 1. In the DOX-treated infant and adult mice, AST was significantly elevated when compared with the control group. In DOX administered infant mice, ALT was significantly elevated, compared with controls. In DOX-treated adult mice, AST/ALT ratio was significantly elevated, compared with the control group. DOX-treated infant mice did not present any changes in their plasma CK-MB nor in their plasma total-CK values. However, in adult mice, both plasma CK-MB and total-CK values were significantly higher than controls.

### 3.3. Histological Changes Were Seen in the Cardiac Structure of Both Mice Groups after DOX Treatment

Histopathological examination of cardiac tissue was performed (Figure 3), using the classical hematoxylin and eosin staining. The results of semi-quantitative analysis of the DOX-treated and control groups are presented in Table 2. The heart tissue from the control groups showed a normal myocardium architecture and a regular cell distribution (I-Control and A-Control). DOX-treated adults exhibited ultrastructural alterations, such as marked interstitial edema, perinuclear vacuolization, disorganization, and degeneration of the myocardium, with extensive DOX-induced loss of myofibril and interstitial inflammatory cell infiltration, as well as some necrotic zones. Infant mice administered with DOX also exhibited ultrastructural alterations, but to a lesser extent when compared to adults (Figure 3 and Table 2).

### 3.4. DOX Treatment Induced Myocardial Fibrosis in Adult Mice

DOX treatment induced a significant increase in collagen deposition in cardiac tissue in adult mice, consistent with interstitial cardiac fibrosis (Figure 4). No significant differences were observed among the infant groups in the percentage of collagen per total section area.

### 3.5. No Significant Changes in the Cardiac Levels of ATP or Glutathione-Related Measurements Were Observed in DOX-Treated Mice of Both Groups

No differences were found in the cardiac levels of tGSH, GSH, GSSG, and GSH/GSSG ratio of the two groups (infant and adult) treated with DOX, compared to controls (Table 3). To understand if DOX had any impact on the cellular energetics on both groups, intracellular ATP was also measured and no statistically relevant differences in ATP levels were observed in both groups of DOX-treated mice, compared to controls.

### 3.6. Protein Carbonylation in Cardiac Lysates Increased in DOX-Treated Infant Mice and Decreased in DOX-Treated Adult Mice

In cardiac lysates, the protein carbonylation increased in the 18.0 mg/kg DOX-treated infant mice compared to control mice (Figure 5A), while DOX-treated adult mice showed a tendency for decreased (*p* = 0.07) levels in comparison with the control group (Figure 5B).

### 3.7. DOX-Treated Infant Mice Showed a Significant Increase in the Heart Levels of Nuclear Factor Erythroid-2 Related Factor 2

DOX-treated infant mice (I-DOX) showed a significant increase in the cardiac levels of Nrf2, compared to control mice (Figure 6A). DOX-treated adult mice (A-DOX) showed no meaningful differences (Figure 6B) in the levels of this transcription factor.

### 3.8. DOX-Treated Mice Showed a Higher Density of Infiltrating M1 Macrophages in the Cardiac Tissue

The number of CD68-positive cells increased in both groups treated with DOX (Figure 7A,B). No meaningful differences were observed in the number of M2 macrophages in the adult (Figure 7D) and infant (Figure 7C) group treated with DOX, when compared to controls.

### 3.9. DOX-Treated Mice Showed a Significant Increase in the Nuclear Factor-ĸB p65 Subunit

The immunohistochemistry data of NF-κB is shown in Figure 8. In the infant group treated with DOX, a significant increase in the levels of NF-κB subunit p65 was observed (I-DOX), just like in the DOX-treated adult group (A-DOX). DOX-treated mice showed positive cytoplasmic and nuclear expressions (brown staining) when compared with the control (I-Control and A-Control). These results were corroborated by semiquantitative analysis, where the DOX-treated infant (Figure 8A) and adult (Figure 8B) groups had a higher number of NF-κB immunopositive cells (*p* < 0.05 and *p* < 0.001, respectively) when compared with the respective control groups.

When total homogenates of cardiac tissue were analyzed via Western blot, DOX led to increased levels of NF-κB p65 in adult mice (Figure 9F), while no differences were observed in the infant mice (Figure 9B). In DOX-treated adult mice, decreased levels of myeloperoxidase (MPO) were seen, compared to control mice (Figure 9H), while no changes were seen in infants (Figure 9D). No differences were observed, neither in the levels of NF-κB p52 (Figure 9C,G) nor of p38 MAPK levels Figure 9A,E) in both groups (Figure 9).

### 3.10. DOX Decreased IL-6 Cardiac Levels, with No Impact on Other Evaluated Cytokines

To determine whether the pro-inflammatory cytokines TNF-α, IL-1β, and IL-6 play key roles in DOX-inflicted cardiotoxicity, Western blotting analysis was performed. DOX led to increased levels of IL-6 in adult mice (Figure 10F), while no differences were observed in the infant mice (Figure 10B). No differences were observed in the levels of IL-1β (Figure 10A,E) and TNF-α (Figure 10C,G) in both groups. However, DOX-treated adult mice showed a tendency toward an increased levels of type 2 TNF receptor (TNFR2) (*p* = 0.07) in comparison with the control group (Figure 10H). This tendency was not observed in infants (Figure 10D).

## 4. Discussion

In the present work, the early myocardial changes caused by DOX were assessed in mice of two age groups and the major findings were: (1) DOX affected the animals’ body weight and food/water consumption; (2) DOX caused a significant elevation of plasma levels of CK-MB, total-CK, AST and AST/ALT ratio in adult mice, which was corroborated by histopathological findings; (3) the ratio between heart weight/brain weight was significantly decreased in both populations; (4) the levels of the macrophage M1 marker and of NF-κB p65 subunit increased in DOX-treated adult animals; (5) no differences were found in the levels of TNF-α, IL-1β, p38 MAPK, and NF-κB p52 in both evaluated populations; (6) in DOX-treated adult animals, the levels of IL-6 significantly decreased; (7) in DOX-treated adult animals, an increase in fibrotic tissue was observed, probably due to the activation of NF-κB transcription factor, while this last was possibly triggered by TNFR2 activation; (8) different responses were seen regarding protein carbonylation after DOX administration in infants and adults; (9) in DOX-treated adult animals, the levels of MPO significantly decreased, perhaps because early damage to the enzyme may lead to less subsequent oxidative stress (namely protein carbonylation), simultaneously blunting the protective Nrf2 activation. On the other hand, in infant mice, increased protein carbonylation leads to the activation of Nrf2, which appears to protect the heart, meaning less damage to the myocardium. We can conclude from the above data that there are significant differences in DOX-induced cardiotoxicity in infants and adults, suggesting distinct molecular and cellular effects that seem related to age.

A significant body weight loss was observed in DOX-treated infant and adult animals, when compared to controls. This loss of weight is likely associated with reduced food consumption since DOX causes loss of appetite, a phenomenon commonly observed in DOX-treated patients [28]. We observed that food and water intake mainly decreased in the following days after the last administration of DOX, so the loss of weight gain can also be an indication of gastrointestinal toxicity. DOX is known to cause gastrointestinal side effects, namely, nausea, vomiting, abdominal pain, gastrointestinal ulceration, and diarrhea [29]. Our data is in accordance with previous findings: after a sub-chronic treatment in rats with weekly subcutaneous injections of DOX over 7 weeks (total cumulative dose: 14 mg/kg), a decrease in body weight at the time of sacrifice was observed [30]; C57BL wild-type Mrp1-null mice treated with DOX [three doses of 3 mg/kg DOX twice a week for 3 weeks (total dose: 18 mg/kg) and five doses of 2 mg/kg DOX twice a week for 5 weeks (total dose: 20 mg/kg, IP)] caused a marked decrease in body weight after treatment. When treatment was discontinued, bodyweight stabilized and began to recover in all DOX-treated animals, as observed in the following 2-week recovery period [28].

Cardiotoxicity was also manifested by a significant elevation of plasma AST and in the AST/ALT ratio in adult mice treated with DOX. The DOX-treated infants had elevated plasma ALT and AST. Other authors have reported increased levels of AST and ALT, after treatment with DOX [31,32,33]. Regarding total-CK and CK–MB, only the DOX-treated adult group had a significant elevation of these enzymes’ activity. In accordance, increased CK-MB plasma levels and electrocardiographic changes have been associated with cardiac tissue lesions [34], which can indicate a serious injury. Moreover, in our study, the heart weight/brain weight ratio decreased significantly in both groups. Previously, rats treated with DOX [total cumulative dose (15 mg/kg)] also showed a significant decrease in the heart weight-to-body weight ratio [35].

In the present work, DOX caused cardiac tissue lesions, as can be seen in the histopathological analysis. DOX-treated adults exhibited ultrastructural alterations, such as marked interstitial edema, perinuclear vacuolization, disorganization, and degeneration of the myocardium. In addition, DOX induced extensive loss of myofibrils and interstitial inflammatory cell infiltration, as well as some necrotic zones. We observed that the inflammatory response in adults treated with DOX was more exuberant and intense, while in infant animals, there is a more controlled and targeted response. Several authors have demonstrated that DOX causes severe pathological lesions in cardiac tissue, namely, cytoplasmic vacuolization, involving the distention of the T-tubules and sarcoplasmic reticulum, perinuclear vacuolization, the congestion of myocardial vessels, disorganization, loss of myofibrillar structure, the swelling and loss of mitochondria, and cellular edema [36,37,38,39,40].

In the present work, we observed that the macrophage M1 marker increased in DOX-treated infant and adult animals, while the NF-κB p65 subunit increased hugely in DOX-treated adult animals. Several in vivo studies have demonstrated that DOX administration leads to inflammation of the cardiac tissue [9,10,11]. DOX causes inflammatory effects in the myocardium through the activation of NF-κB [13,14,15]. Nevertheless, in our work, no differences were found in the levels of IL-1β, p38 MAPK, NF-κB p52 and TNF-α in both groups. Conversely, in other studies, DOX elicited a series of inflammatory responses within the myocardium by upregulating NF-κB p65 and inducing the subsequent release of several proinflammatory cytokines implicated in cardiac pathogenesis and apoptosis, such as IL-1β, IL-6, TNF-α, and p38 MAPK [9,13,40,41,42], which was not in full accordance with our results. Zhu et al. observed an increase in serum levels of IL-1β in Balb/c mice treated with DOX (18 mg/kg, IP) [9]. Wistar albino rats that received 2.5 mg/kg of DOX (IP) thrice weekly (15 mg/kg cumulative dose) suffered an increase in cardiac inflammatory markers, including TNF-α and IL-1β and caspase-3 [40]. DOX increased inflammatory cytokines in the heart, including TNF-α and NF-κB, in Male Wistar rats injected with DOX (IP) (13 mg/kg cumulative dose in 6 injections over 2 weeks) [41] and after a single IP injection of DOX (15 mg/kg) [42]. H9c2 cells exposed to DOX showed a stimulation of the inflammatory response, namely, an increase in the levels of IL-1β, IL-6, TNF-α, as well as overexpression of phosphorylated p38 MAPK and phosphorylation of the NF-κB p65 subunit [13]. Another study, this time in vitro, showed that DOX caused a significant increase in the production of IL-8, IL-6, and IL-1β, suggesting an important role of inflammation in DOX-induced cardiotoxicity [43]. According to the recent literature, galectin-3 and growth stimulation-expressed gene 2 (ST2) might be used as a diagnostic or prognostic biomarkers for HF [44]. Galectin-3 is involved in ROS production and is positively correlated with markers of oxidative stress in some conditions [45,46,47]. In male Sprague-Dawley rats injected with a single dose of DOX (15 mg/kg, IP), the authors observed that Galectin-3 mediated DOX-induced cardiotoxicity and that Galectin-3 inhibition with modified citrus pectin attenuated DOX-induced cardiac dysfunction by upregulating the expression of antioxidant peroxiredoxin-4 and reducing myocardial oxidative stress [48]. On the other hand, ST2 encodes a transmembrane protein for the interleukin-1 receptor family (ST2L), and a truncated soluble form of ST2 (sST2), secreted from cardiac myocytes and cardiac fibroblasts, triggered mechanical cardiac strain [49,50]. ST2L and sST2 bind to interleukin-33 (IL-33), a cytokine produced by cardiac fibroblasts, causes two opposing effects: (1) the binding of IL-33 and ST2L results in cardioprotective effects against hypertrophic remodeling and myocardial fibrosis by antagonizing angiotensin-II signaling and promoting anti-apoptotic factors, respectively; (2) sST2 bound to IL-33 acts as a decoy receptor, and inhibits the protective effects of IL-33 and ST2L in cardiac myocytes [51,52]. Moreover, it has been demonstrated that IL-33 inhibited DOX-induced ROS, prevented apoptosis signal-regulating kinase 1 and c-Jun N-terminal kinase phosphorylation, and attenuated the DOX-induced myocyte apoptosis in cardiomyocytes derived of c-Jun N-terminal kinase-deficient C57BL/6 mice [53]. Nonetheless, in our work, DOX-treated adult mice showed a tendency toward increased TNFR2, perhaps because this receptor possibly activates NF-κB [54]. In relation to IL-6, its expression was decreased in adult mice treated with DOX, which is in accordance with our previous work with mitoxantrone [18], an anticancer drug that shares similar clinical cardiotoxicity with DOX. In another work, the reduction of IL-6 levels and insulin-like growth factor (IGF)-1, accompanied by the simultaneous enhanced production of TNF-α, may have critically contributed to the development of the HF phenotype observed [55]. Thus, we believe that, in adults, TNFR2 activation leads to NF-κB activation, which may result in cardiac damage, as we will report next.

DOX treatment induced a significant increase in collagen deposition in the cardiac tissue of adult mice, consistent with interstitial cardiac fibrosis. These results indicate that in the early stages of DOX-induced cardiotoxicity, the increase in interstitial fibrosis occurs due to activation of the NF-κB transcription factor, which is also corroborated by a previous study [56]. Tanaka et al. observed that low-dose DOX might induce reactive fibrosis and that reactive fibrosis preceded DOX-induced HF through inflammation at the initial stage [57]. Another study observed a significant increase in interstitial fibrosis in animals treated with DOX (1 mg/kg IV twice a week for eight weeks, followed by a drug-free period of 1 week) suggesting a role for NF-κB in that DOX-induced fibrosis [56]. In that paper, the authors also provided an free drug period after DOX treatment to allow the heart to respond.

Protein carbonylation is a frequent type of ROS-induced protein modification, being considered irreversible, and aims to induce protein degradation [58]. A tendency toward a decrease in carbonylated proteins was observed in DOX-treated adult mice, suggesting activation of proteasomal degradation of oxidatively modified proteins and stress response [59], or other mechanisms that still need investigation. Conversely, DOX-treated infant mice had increased levels of carbonylated proteins, indicating the early induction of oxidative stress and protein damage. Several studies demonstrated that DOX causes an increase in protein carbonyl groups in cardiac tissue [36,37,60]. A significant increase in total cardiac protein carbonylation was measured in rats that were treated with DOX (10 mg/kg IP) and euthanized two weeks after treatment [60]. The same work showed that 500 nM of DOX induces carbonylation and the degradation of cardiac myosin-binding protein C, and cytotoxicity in HL-1 cells at 24 h [60]. Moreover, there was a substantial increase in the concentration of carbonyl groups in the cardiac tissue of DOX-injected rats, which received seven weekly subcutaneous injections of DOX (2 mg/kg) [36,37].

MPO catalyzes the production of the powerful oxidant hypochlorous acid, using H_2_O_2_ and chloride as substrates [61]. MPO is expressed in cells of myeloid origin and is capable of the enzymatic conversion of topoisomerase II poisons, namely, DOX into species with greater DNA-damaging activity [62]. In our study, the levels of MPO significantly decreased in adult mice treated with DOX, with no changes suggested in the infant group. In the past, MPO inhibition seemed to partially protect myeloid precursors from topoisomerase II poison-mediated DNA damage [62]. The decrease in MPO levels in adult animals treated with DOX in our work suggests that there is less oxidative stress in that group at the time of sacrifice. On the other hand, in infant mice treated with DOX, no differences were found in the levels of MPO when compared to control mice. This result may mean that MPO can still contribute to the oxidative stress elicited by DOX [63], as observed in the results of protein carbonylation and the possible activation of Nrf2, as will be discussed next in infant mice.

In the present work, the levels of Nrf2 were not changed in DOX-treated adult mice. On the other hand, the levels of Nrf2 significantly increased in DOX-treated infant mice. Nrf2 inhibits NF-κB activation, by preventing the degradation of IκBα, and increases HO-1 expression and also the expression of other antioxidant defenses, which neutralize ROS [64,65]. Actually, the total expression of cardiac NF-κB did not change in infants (although we saw scattered but significant increases by the immunohistochemistry method), where Nrf2 was activated. Moreover, the activation of Nrf2 has been reported to favor the reduction of pro-inflammatory cytokine levels in cells [64]. Several studies have investigated the role of Nrf2 in DOX-induced cardiotoxicity, both in vivo and in vitro [66]: (1) Nrf2 deficiency increased cardiotoxicity and cardiac dysfunction in DOX-treated mice (a single IP injection of 25 mg/kg), which highlights the potential of Nrf2 as a new therapeutic strategy to attenuate DOX-induced cardiotoxicity and cardiomyopathy [66]; (2) several candidates against DOX-induced cardiotoxicity have been studied focusing on Nrf2, namely, naringenin-7-O-glucoside [67], sulforaphane [68], and tert-butylhydroquinone [69]; (3) Nrf2 mediated the protective effect of α-linolenic acid [70] and tanshinone IIA [71] on DOX-induced cardiotoxicity, through the suppression of oxidative stress. The transcription factor Nrf2 is critical in the regulation of a large array of antioxidant proteins, including GSH-synthesizing enzymes [72]. In our study, the cardiac levels of the tGSH, GSSG and GSH/GSSG ratios were not different among groups at the time of sacrifice, but other genes may have been activated, or those changes may occur later on. Still, infant mice seem to be more protected from damage caused by DOX, as they appear to have activated antioxidant defenses, namely, Nrf2, that can prevent inflammation and further histological damage. Although it may seem counterproductive, the lower levels of MPO seen in the adults may blunt Nrf2 activation, while changes in protein carbonylation could activate a protective effect in infants.

## 5. Conclusions

Our data indicate that in adult mice, the increase in interstitial fibrosis occurs due to activation of the NF-κB transcription factor, possibly by TNFR2 activation. However, the decrease in the levels of MPO observed in adult animals may suggest early damage to the enzyme related to DOX exposure, leading to subsequent lower oxidative stress (namely, protein carbonylation), while simultaneously blunting the protective Nrf2 activation. Notwithstanding this, DOX leads to a reduction in the levels of IL-6 in adult mice, which may contribute to the development of the HF phenotype later on. Conversely, in infant mice, the activation of Nrf2 seems to be cardioprotective in this group, where less myocardial damage is seen. Concerning infant mice, no differences were found in the levels of MPO, suggesting that this enzyme may contribute to increased protein carbonylation, while leading to the activation of Nrf2. The Nrf2 pathway activation may prevent inflammation by inhibiting general NF-κB activation in that group, at least in the total homogenate. To conclude, infant mice seem to be more protected from damage caused by DOX, as they appear to have activated antioxidant defenses in contrast to what was observed in adult mice.

## Figures and Tables

**Figure 1 biomolecules-11-01725-f001:**
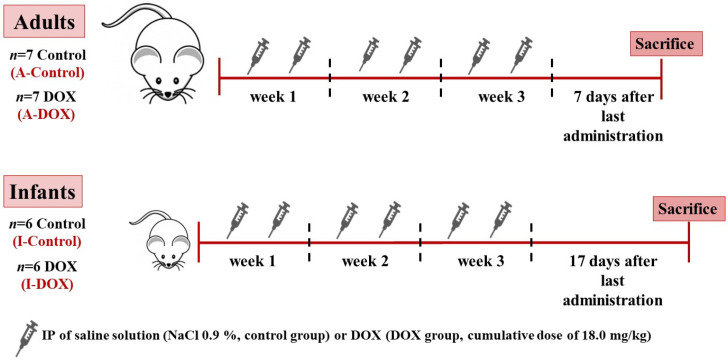
Schematic representation of the methodology used. Over three weeks, mice were biweekly injected intraperitoneally (IP) with saline solution (control groups) or DOX (total cumulative dose: 18.0 mg/kg, DOX groups, which corresponds in humans to 100.6 mg/m^2^ for infants and 108.9 mg/m^2^ for adults).

**Figure 2 biomolecules-11-01725-f002:**
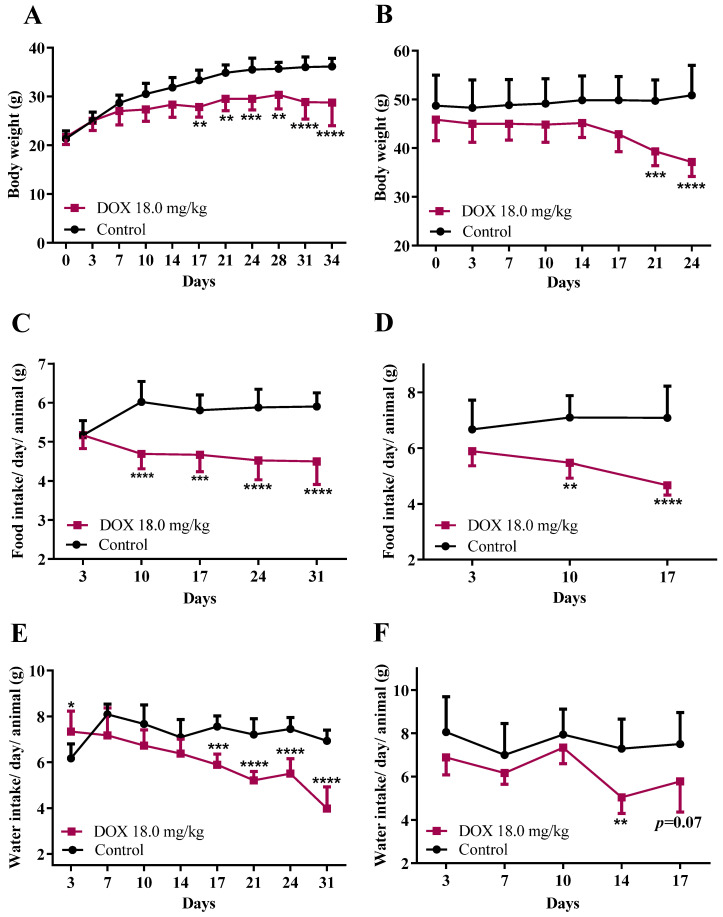
Body weight, food and water intake in infant and adult mice exposed to a cumulative dose of 18.0 mg/kg DOX (pink squares) compared with controls (black circles). Body weight in (**A**) infant and (**B**) adult mice. Food consumption of (**C**) infant and (**D**) adult mice. Water consumption of (**E**) infant and (**F**) adult mice. Results are presented in grams (g) of food intake/day/weight of animal, mL of water intake/day/weight of the animal or g of body weight, and as means ± standard deviation (SD), from 6 infant or 7 adult mice per group. Statistical comparisons were made using a two-way ANOVA followed by the Sidak’s *post hoc* test (** *p* < 0.01, *** *p* < 0.001, and **** *p* < 0.0001, DOX 18.0 mg/kg vs. control).

**Figure 3 biomolecules-11-01725-f003:**
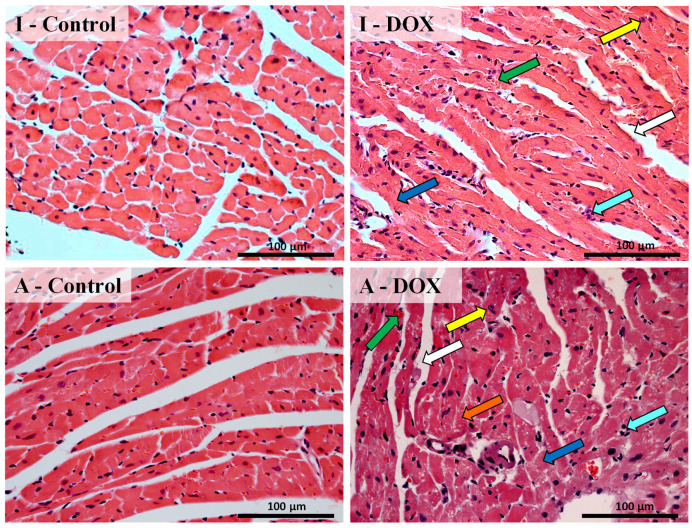
Cardiac histopathology, using light microscopy, of infant DOX-treated (**I-DOX**) and control mice (**I-Control**) and of adult mice exposed to a cumulative dose of 18.0 mg/kg DOX (**A-DOX**) and respective controls (**A-Control**), assessed by hematoxylin and eosin staining. Representative light micrograph from: infant mice controls, showing normal morphology and structure; infant mice given a cumulative dose of 18.0 mg/kg DOX; control adult mice, showing normal morphology and structure; and adult mice given a cumulative dose of 18.0 mg/kg DOX. DOX-treated groups presented large and uncondensed nucleus (yellow arrow), extensive loss of myofibril (white arrow), vacuolization (orange arrow), inflammatory infiltration, (cyan arrow) and vascular congestion (green arrow). Necrotic zones (blue arrow) are evident. Scale bar = 100 µm, *n* = 3. Images were taken at 40× magnification.

**Figure 4 biomolecules-11-01725-f004:**
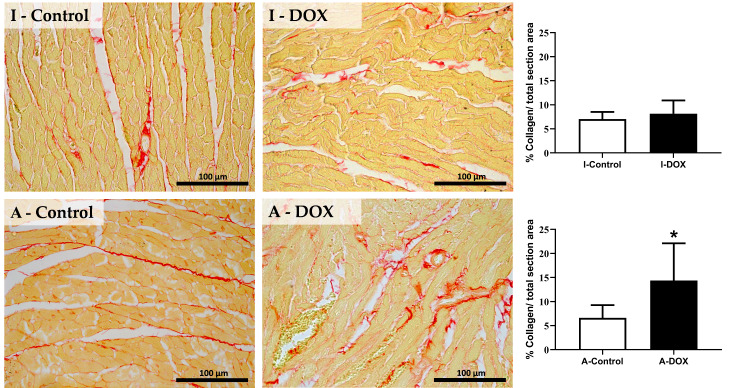
Sirius red staining assessed by light microscopy and semi-quantitative analysis of fibrosis in the heart of infant DOX-treated (**I-DOX**) and control mice (**I-Control**), and adult mice exposed to a cumulative dose of 18.0 mg/kg DOX (**A-DOX**) and respective controls (**A-Control**). Images were taken at 40× magnification. Scale bar = 100 µm. The results of collagen content red staining are expressed as a percentage of collagen per total section area. Statistical comparisons were made using the Mann–Whitney test: * *p* < 0.05, DOX vs. control.

**Figure 5 biomolecules-11-01725-f005:**
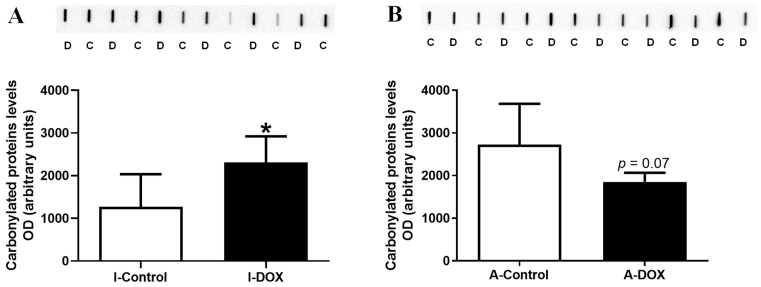
Protein carbonylation cardiac lysate content evaluated by slot blot in (**A**) infant DOX-treated (I-DOX) and control mice (I-Control), and (**B**) adult mice exposed to a cumulative dose of 18.0 mg/kg DOX (A-DOX) and respective controls (A-Control). Values are expressed as mean ± SD and were obtained from 6 infant or 7 adult animals from each treatment group. Statistical comparisons were made using Student’s *t*-test: * *p* < 0.05, DOX 18 mg/kg vs. control. OD: optic density; D: DOX and C: Control.

**Figure 6 biomolecules-11-01725-f006:**
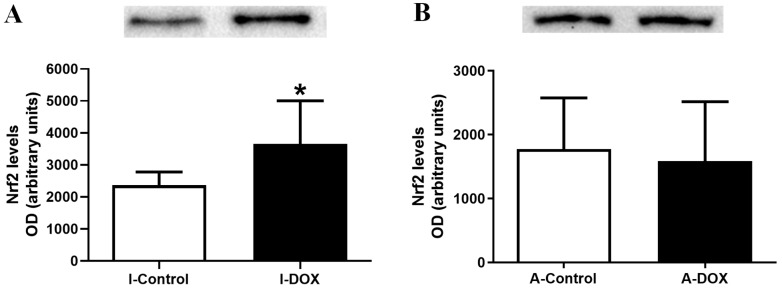
Nuclear factor erythroid-2 related factor 2 (Nrf2) (97 kDa) levels in the cardiac tissue evaluated by Western blotting, in (**A**) infant DOX-treated (I-DOX) and control mice (I-Control), and (**B**) adult mice exposed to a cumulative dose of 18.0 mg/kg DOX (A-DOX) and respective controls (A-Control). Values are expressed as mean ± SD and were obtained from 6 infant or 7 adult animals from each treatment group. Statistical comparisons were made using Student’s *t*-test: * *p* < 0.05, DOX vs. control. OD: optic density. Protein loading was confirmed by the Ponceau S staining (Appendix A).

**Figure 7 biomolecules-11-01725-f007:**
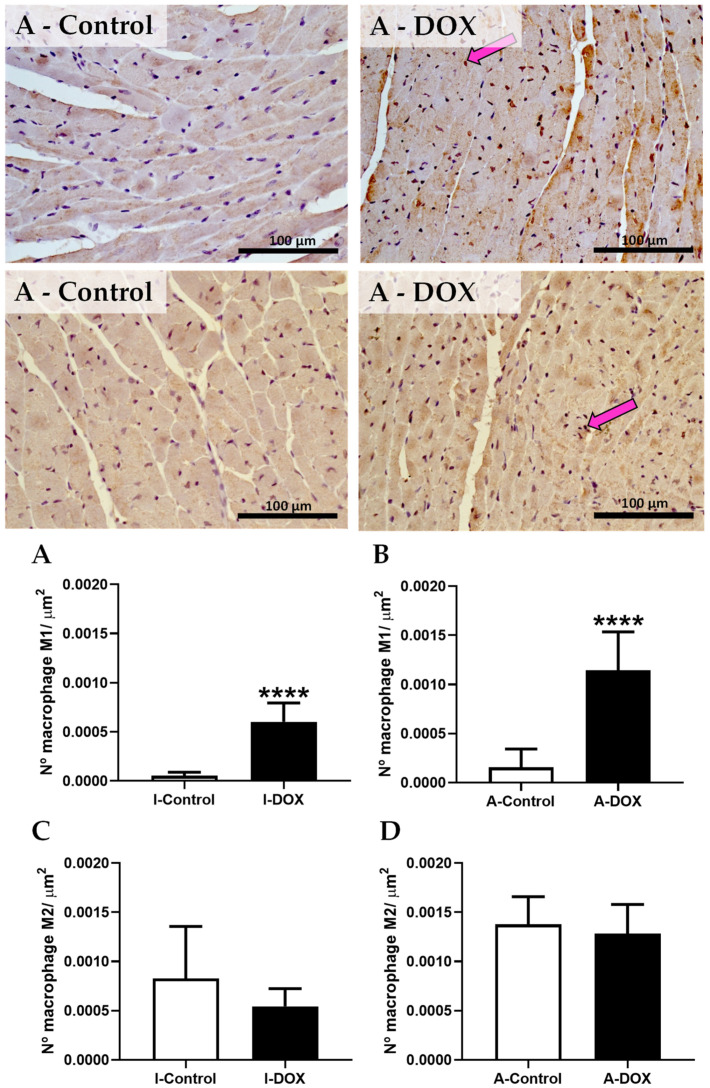
Representative photomicrographs of immunohistochemistry done in cardiac tissue of DOX-treated adult (A-DOX) and respective controls (A-Control), of CD68-positive cells (a marker for the macrophage M1) (on the top line) and of CD206 positive cells (a marker for the macrophage M2) (the bottom histologic figures). Positive stains are indicated by pink arrows. Number of cells staining positive for activated macrophages marked as M1 in adults (**A**) and infants (**B**). Number of cells staining positive for activated macrophages marked as M2 in the heart of (**C**) infant DOX-treated (I-DOX) and control mice (I-Control), and (**D**) adult mice exposed to a cumulative dose of 18.0 mg/kg DOX (A-DOX) and respective controls (A-Control). Results were expressed as the number of positive cells per area (µm^2^) as mean ± SD. Results were obtained from three animals from each treatment group (six random fields per *n*). Statistical comparisons were made, using Student’s *t*-test for macrophage M2 in the infant group and by the Mann–Whitney test for all other analyses: **** *p* < 0.001, DOX vs. control. Scale bar = 100 µm, *n* = 3. Images taken at 40×.

**Figure 8 biomolecules-11-01725-f008:**
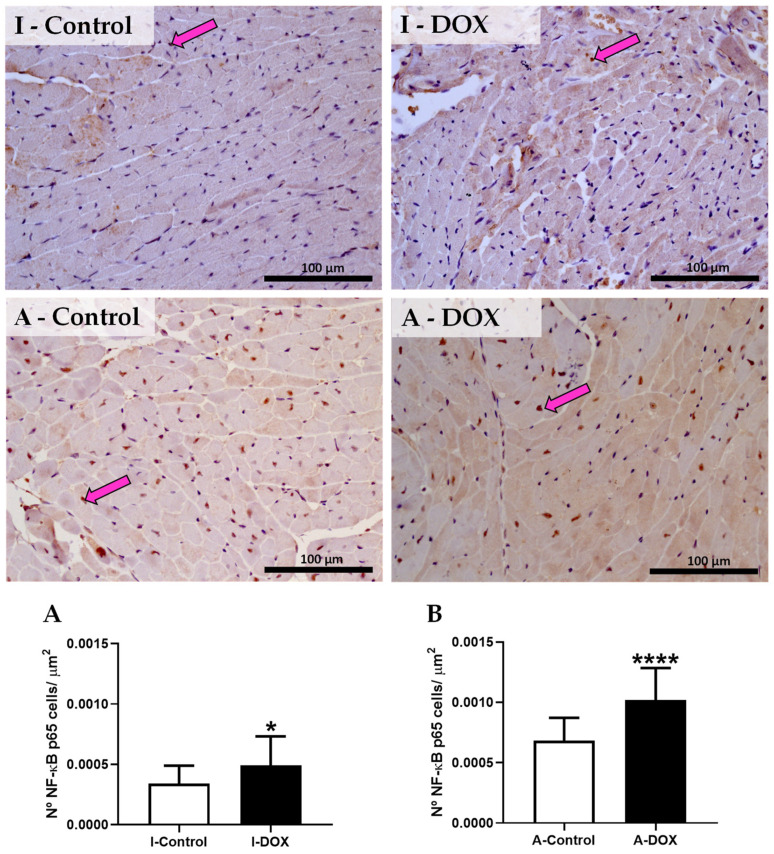
Representative photomicrographs of the immunohistochemistry determination of nuclear factor kappa B (NF-κB) in the cardiomyocytes-like cells from infant DOX-treated (I-DOX) and control mice (I-Control), and adult mice exposed to a cumulative dose of 18.0 mg/kg DOX (A-DOX) and the respective controls (A-Control). Number of cells staining as positive, indicated by pink arrows for the activated NF-κB of the heart of DOX-treated and control groups, in (**A**) infant and (**B**) adult animals. The results were expressed according to the number of positive cells per area (µm^2^) as mean ± SD. Results were obtained from three animals from each treatment group. Statistical comparisons were made using the Mann–Whitney test: * *p* < 0.05, **** *p* < 0.001, DOX vs. control. Scale bar = 100 µm, *n* = 3 (six random fields per *n*). Images were taken at 40×.

**Figure 9 biomolecules-11-01725-f009:**
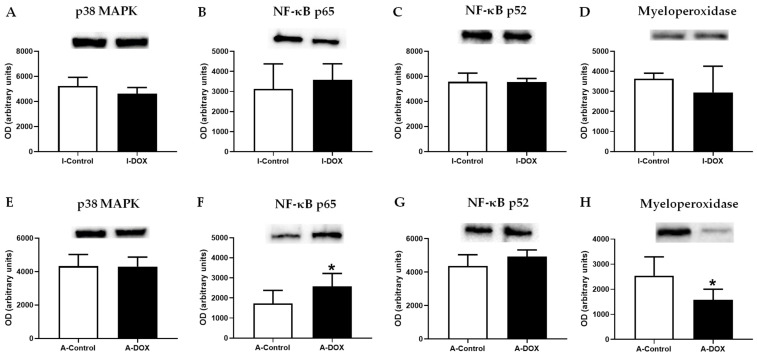
(**A**,**E**) p38 MAP Kinase (p38 MAPK) (40 kDa), (**B**,**F**) NF-κB p65 (60 kDa), (**C**,**G**) NF-κB p52 (50 kDa), (**D**,**H**) myeloperoxidase (MPO) (48 kDa) levels of cardiac tissue evaluated by Western blotting, from infant DOX-treated (I-DOX) and control mice (I-Control), and adult mice exposed to a cumulative dose of 18.0 mg/kg DOX (A-DOX) and respective controls (A-Control). Values are expressed as mean ± SD and were obtained from 6 (infant) and 6–7 (adult) animals from each treatment group. Statistical comparisons were made using Student’s *t*-test: * *p* < 0.05, DOX vs. control. OD: optic density. Protein loading was confirmed by Ponceau S staining (Appendix A).

**Figure 10 biomolecules-11-01725-f010:**
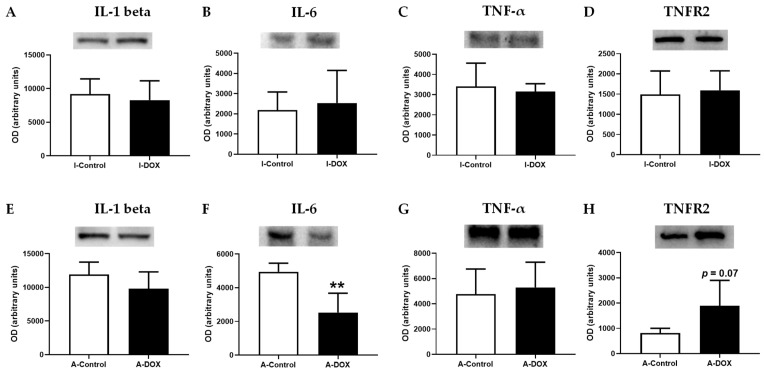
(**A**,**E**) Interleukin-1 beta (IL-1 beta) (35 kDa), (**B**,**F**) interleukin-6 (IL-6) (24 kDa), (**C**,**G**) tumor necrosis factor- α (TNF-α) (25 kDa), (**D**,**H**) type 2 TNF receptor (TNFR2) (75 kDa) levels in cardiac tissue evaluated by Western blotting, in infant DOX-treated (I-DOX) and control mice (I-Control), and adult mice exposed to a cumulative dose of 18.0 mg/kg DOX (A-DOX) and respective controls (A-Control). Values are expressed as mean ± SD and were obtained from 6 (infant) and 6–7 (adult) animals from each treatment group. Statistical comparisons were made using Student’s *t*-test: ** *p* < 0.01, DOX vs. control. OD: optic density. Protein loading was confirmed by the Ponceau S staining (Appendix A).

**Table 1 biomolecules-11-01725-t001:** Plasma biomarkers and heart weight-to-brain weight ratio after DOX (18.0 mg/kg cumulative dose).

Plasma	I-Control	I-DOX	A-Control	A-DOX
**AST(U/L)**	37.67 ± 6.02	**58.40 ± 12.10 ***	35.29 ± 14.45	**86.25 ± 26.32 ****
**ALT (U/L)**	9.33 ± 4.27	**26.75 ± 3.30 ***	15.86 ± 8.65	28.17 ± 19.10
**AST/ALT ratio**	4.81 ± 2.37	2.76 ± 1.29	2.33 ± 0.80	**9.63 ± 5.79 ***
**CK-MB (U/L)**	55.83 ± 22.27	69.40 ± 60.91	31.67 ± 19.56	**154.5 ± 105.7 ****
**Total-CK (U/L)**	43.33 ± 14.42	101.8 ± 111.2	22.83 ± 11.48	**401.7 ± 369.7 ****
**Heart**	**I-Control**	**I-DOX**	**A-Control**	**A-DOX**
**Heart weight/** **brain weight ratio (%)**	0.38 ± 0.03	**0.29 ± 0.04 ****	0.60 ± 0.08	**0.42 ± 0.17 ****

Plasma biomarkers and heart weight-to-brain weight ratio in infant DOX-treated (I-DOX) and control mice (I-Control), and adult mice exposed to a cumulative dose of 18.0 mg/kg DOX (A-DOX) and respective controls (A-Control). Data of creatine-kinase-MB (CK-MB), total creatine-kinase (total CK), AST and ALT levels, in units per liter (U/L), and data of the heart weight-to-brain weight ratio are presented as means ± SD. Data were obtained from 6 infant or 7 adult animals from each group. Statistical comparisons were made in each age group using the Mann–Whitney test for all experiments. * *p* < 0.05, ** *p* < 0.01, DOX vs. control.

**Table 2 biomolecules-11-01725-t002:** Semi-quantitative analysis of the morphological parameters (cellular degeneration, necrotic zones, interstitial inflammatory cell infiltration and loss of tissue organization) in the heart of DOX-treated and controls, in adult and infant animals.

Hematoxylin-Eosin Staining	I-Control	I-DOX	A-Control	A-DOX
**Cellular degeneration**	0.07 ± 0.25	**0.57 ± 0.63 *****	0.02 ± 0.13	**2.45 ± 0.50 ******
**Necrotic zones**	0.00 ± 0.00	**0.30 ± 0.47 ****	0.00 ± 0.00	**1.40 ± 0.50 ******
**Interstitial inflammatory cell infiltration**	0.08 ± 0.27	**0.53 ± 0.51 *****	0.05 ± 0.22	**1.26 ± 0.44 ******
**Loss of tissue organization**	0.00 ± 0.00	0.00 ± 0.00	0.00 ± 0.00	**0.38 ± 0.50 *****

Semi-quantitative analysis of the morphological parameters in the heart of infant DOX-treated (I-DOX) and control mice (I-Control), and adult mice exposed to a cumulative dose of 18.0 mg/kg DOX (A-DOX) and respective controls (A-Control). Results, given in scores, are presented as means ± SD and were obtained from three animals from each treatment group (ten random fields per animal). Statistical comparisons were made using the Mann–Whitney test between the same age groups: ** *p* < 0.01, *** *p* < 0.001, **** *p* < 0.0001, DOX vs. control.

**Table 3 biomolecules-11-01725-t003:** Biochemical cardiac parameters of the DOX-treated and control mice.

	I-Control	I-DOX	A-Control	A-DOX
**tGSH (nmol/mg protein)**	3.700 ± 0.913	3.864 ± 1.872	3.853 ± 1.404	2.843 ± 1.556
**GSSG (nmol/mg protein)**	0.827 ± 0.258	0.693 ± 0.400	0.871 ± 0.244	0.503 ± 0.410
**GSH (nmol/mg protein)**	2.046 ± 0.846	2.478 ± 1.206	2.111 ± 1.213	1.837 ± 1.091
**GSH/GSSG ratio**	2.107 ± 1.306	3.880 ± 2.234	2.465 ± 1.159	5.143 ± 4.009
**ATP (nmol/mg protein)**	0.484 ± 0.086	0.673 ± 0.357	0.382 ± 0.346	0.322 ± 0.074

Biochemical cardiac parameters of infant DOX-treated (I-DOX) and control mice (I-Control), and adult mice exposed to a cumulative dose of 18.0 mg/kg DOX (A-DOX) and respective controls (A-Control). Results, in nmol/mg or GSH/GSSG ratio, are presented as means ± SD and were obtained from 6 infant and 7 adult animals from each treatment group. Statistical analyses were made using the Mann–Whitney test.

## Data Availability

The data presented in this study are available on request from the corresponding authors.

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
