# Peer review of "Role of Inflammation and Redox Status on Doxorubicin-Induced Cardiotoxicity in Infant and Adult CD-1 Male Mice"

_biomolecules, 2021, doi:10.3390/biom11111725_

Round 1

Reviewer 1 Report

To:

Editorial Board

Biomolecules

Title: “Role of inflammation and redox status on doxorubicin-induced 2 cardiotoxicity in infant and adult CD-1 male mice”

Dear Editor,

I read this paper and I think that:

  • I cannot understand why the authors divided mice according to age. The translation of this paper into human clinical practice should take into account age, rather than basic researches. Please discuss such a point.
  • The role of IL-33 and ST2 as well as galectin could be of interest in such a context. The authors can discuss such a point in relation to literature.

Author Response

COMMENTS OF REVIEWER 1: “Dear Editor, I read this paper and I think that: I cannot understand why the authors divided mice according to age. The translation of this paper into human clinical practice should take into account age, rather than basic researches. Please discuss such a point.”

AUTHOR’S RESPONSES TO REVIEWER 1: We thank you for the reviewer’s comment. Cardiotoxicity in cancer pediatric patients has been given a lot of attention, because of their life expectancy, higher susceptibility risk and because of age-related differences; moreover, clinical management and preclinical studies must consider unique challenges faced by the pediatric population [1]. Actually, pediatric cancer survivors who do develop cardiac dysfunction are often afflicted in their 30s and 40s, their peak of active life, meaning that they have to endure a life-threatening and debilitating disease at a much younger age, affecting their quality of life and increase the cost of care [1]. Thorough the work we tried to highlight that age had impact on the cardiotoxicity observed, to unravel the molecular mechanisms responsible for DOX-induced cardiotoxicity or biomarkers at two different ages. 

For that, we added in introduction of manuscript:  

Line 87-91 page 3: While DOX exposure has been assumed to have similar acute cellular effects in the heart, regardless of age, there is mounting evidence that highlights the significant clinical differences between pediatric and adult DOX-induced cardiotoxicity. Distinct molecular and cellular effects could influence everything from mechanisms of acute damage, injury repair, and pathologic progression to overt cardiac disease [7].

We added in discussion of manuscript:  

Line 659-660 page 18: “We can conclude from the above data that there are significant differences between infant and adult DOX-induced cardiotoxicity, suggesting distinct molecular and cellular effects that seem related to age.”

COMMENTS OF REVIEWER 1: “The role of IL-33 and ST2 as well as galectin could be of interest in such a context. The authors can discuss such a point in relation to literature.”

AUTHOR’S RESPONSES TO REVIEWER 1: We acknowledge the reviewer’s point of view. We add these subjects to the discussion of manuscript to highlight the role of IL-33, ST2 and galectin on this context:

Line 761-849 page 19-20:According to the recent literature, galectin-3 and growth stimulation expressed gene 2 (ST2) might be used as a diagnostic or prognostic biomarker for HF, namely aimed to be used as a predictor of death and hospitalization [44]. Galectin-3 has been involved in ROS production and positively correlated with markers of oxidative stress in some conditions [45-47]. In male Sprague-Dawley rats, injected with a single dose of DOX (15 mg/kg, IP), the authors observed that Galectin-3 mediated DOX-induced cardiotoxicity and Galectin-3 inhibition with modified citrus pectin attenuated DOX-induced cardiac dysfunction by upregulating the expression of antioxidant peroxiredoxin-4 and reduced myocardial oxidative stress [48]. ST2 encodes a transmembrane protein for interleukin-1 receptor family (ST2L), and a truncated soluble form of ST2 (sST2) secreted from cardiac myocytes and cardiac fibroblasts, triggered mechanical cardiac strain [49, 50]. ST2L and sST2 bind to interleukin-33 (IL-33), a cytokine produced by cardiac fibroblasts, causing two opposing effects: 1) binding of IL-33 and ST2L results in cardioprotective effects against hypertrophic remodeling and myocardial fibrosis by antagonizing angiotensin-II signaling and promoting anti-apoptotic factors, respectively; (2) sST2 bound to IL-33 acts as a decoy receptor, and inhibits the protective effects of IL-33 and ST2L in cardiac myocytes [51, 52]. Moreover, it has been demonstrated that IL-33 inhibited the DOX-induced ROS, prevented apoptosis signal-regulating kinase 1 and c-Jun N-terminal kinase phosphorylation and attenuated the DOX-induced myocyte apoptosis in cardiomyocytes derived from c-Jun N-terminal kinase deficient C57BL/6 mice [53].”

Reviewer 2 Report

Reaserch concerns role of inflammation and redox status on doxorubicin-induced 2 cardiotoxicity in infant and adult CD-1 male mice.

What means 400-550 mg/m2? Strange description of dosage. What was the rationale for the doses used

The article needs a lot of editing...sentences at the end of the page like lines 211, 317....
secondly, the size of panels in the figures - graphs of figures 2 or 3 are large while the panels of figures 7 and 8 are sometimes difficult to read - it would be useful to standardize them somehow 

of more important things in the text throughout the manuscript the word expression is used incorrectly....if the authors studied the expression of genes real time PCR we can say so, while I see mainly Wester-blotting analysis and here we have only the level of a given protein - these are two different things (the abstract itself is full of them)

I am a bit puzzled by the procedure of the experiment - why one group had 14 and the other 12 individuals (not equally) and why the experiment was conducted only on males? Often experiments on animals concern only one sex and then we do not have a full picture

second, the authors develop the procedure on some previously developed scale. My question is how many papers have used this scale? Is it commonly used for this type of experiments and if it is used, is it calculated on the basis of the average of several experimenters? my understanding is that all scales are inherently a somewhat subjective assessment

Carbonylation of proteins as a measure of ROS is the first time I've encountered such an assessment...would the authors consider confirming with another method. it doesn't appeal to me....there is a slot dot measurement where by design of the method we have no loading counter

While we are on the subject of loading control, the same is true of Ponceau staining ... it is rather used classically to check whether the transfer was successful ... most often a measurement of the level of a reference protein e.g. actin is added and during densitometric analysis it is recalculated against it and this should be on the figure

I honestly don't understand what the authors are presenting on the y-axis in their analysis of western blotting??? not to mention the fact that in the figures it should be tested stripe and control of loading.... captions of which protein level we are testing should be in the title and not a cluster of graphs and it is on the y-axis or in the caption of the figure that I have to look for what was tested here, in addition, some photos have terrible pixelation.... I would like to see whole membranes. the quality of some photos is unacceptable and I can not even say which ones because figures 9 and 10 do not have a division into panels

discussion as if written backwards - pointing out what new has been discovered as a form of summary is ok but as a discussion starter it simply does not look like it

The work is very similar in structure to the quoted item 17 written by the authors this year - a different compound was used... I have an impression that a whole series will be created or maybe already created. What is the authors' explanation for the submission to biomolecules? I am not sure it is right fit

Author Response

COMMENTS OF REVIEWER 2: “Research concerns role of inflammation and redox status on doxorubicin-induced cardiotoxicity in infant and adult CD-1 male mice. What means 400-550 mg/m2? Strange description of dosage. What was the rationale for the doses used.

AUTHOR’S RESPONSES TO REVIEWER 2: We thank the reviewer’s comments and criticisms to increase the manuscripts’ impact. Usually, in the clinical practice, DOX is administrated according to the patient’s surface area; the usual administration schedule is 60-75 mg/m2 every 3 weeks, depending on the tumour, but should not surpass the total lifetime cumulative dose of 400-550 mg/m2 [2] due to the increased risk of cardiotoxicity. On the other hand, herein we used mg/kg regarding animal weight to compare to other pre-clinical studies published before.

We made the conversion of dose to weight of animal [3-5], considering the Km factor for calculating the dose as a function of the animal's age, as well as the animal weights and average human weight. We applied the allomeric scaling for converting human dose to animal dose:

  • Human dose (mg/kg) = animal dose (mg/kg) x (animal body weight/ human body weight) (1/4)
  • Human dose (mg/kg) x Km factor = Human dose (mg/m2)

Human Weight (kg)

Animal Weight (kg)

Km factor

Animal dose (mg/kg)

Human dose (mg/kg)

Human dose mg/m2

Adult

70

0.050

37

18

2.94

108.9

Infant

8

0.020

25

18

4.02

100.6

Thus, doses used in this manuscript have clinical significance as we stated in the methods section. This total cumulative dose of DOX corresponds approximately in human to 100.6 mg/m2 for infants and 108.9 mg/m2 for adults [3-5]. However, there are data in the literature that show that in cumulative doses below the maximum total lifetime cumulative dose, cardiotoxicity is already observed [2].

We have in the introduction section, the following information:

Line 76-77 page 3: “It has been estimated that 3 % of patients develop DOX-related HF after a cumulative dose of 400 mg/m2, 7 % after cumulative dose of 550 mg/m2, and 18 % at 700 mg/m2 [3].”

COMMENTS OF REVIEWER 2: “The article needs a lot of editing...sentences at the end of the page like lines 211, 317....”  

AUTHOR’S RESPONSES TO REVIEWER 2: We proceeded with several changes in the text as suggested and can be observed in the track changes manuscript.

COMMENTS OF REVIEWER 2: “secondly, the size of panels in the figures - graphs of figures 2 or 3 are large while the panels of figures 7 and 8 are sometimes difficult to read - it would be useful to standardize them somehow”

AUTHOR’S RESPONSES TO REVIEWER 2: To make figure 7 and 8 clearer and more similar among each other, we placed the images and graphics in a horizontal fashion in the manuscript.

COMMENTS OF REVIEWER 2: of more important things in the text throughout the manuscript the word expression is used incorrectly....if the authors studied the expression of genes real time PCR we can say so, while I see mainly Wester-blotting analysis and here we have only the level of a given protein - these are two different things (the abstract itself is full of them)”

AUTHOR’S RESPONSES TO REVIEWER 2: According to the reviewer´s suggestion, we changed in the manuscript the following word “expression” to “levels”, whenever protein levels were concerned.

COMMENTS OF REVIEWER 2: “I am a bit puzzled by the procedure of the experiment - why one group had 14 and the other 12 individuals (not equally) and why the experiment was conducted only on males? Often experiments on animals concern only one sex and then we do not have a full picture”

AUTHOR’S RESPONSES TO REVIEWER 2: We acknowledge the reviewer’s point of view. The infant mice was raised in the vivarium, because our supplier does not provide animals at such an early age, thus we worked with the male litter born at that time, to make sure they were similar in age and the experiment could be performed. In the adults, we could increase the subjects per group. To use a higher number of animals in those analyses, we would need more samples and it would be necessary to replicate the entire protocol. Considering the 3Rs for animal models, and as our protocol causes suffering to the animals, it was not correct ethically neither practically to repeat the in vivo protocol. Thus, the non-equality in the number of animals per group (n = 14 adults and n = 12 infants) is due to the availability of animals at the time of the protocol, considering sufficient, but not excessive, sample size to ensure statistical significance of biologically relevant effects. Moreover, regarding the second aspect raised by the reviewer, most in vivo studies use males, so it is easier to make a correlation to the data described in the literature. To use females, or if we had chosen to address a cancer model of breast cancer for instances, we would have to choose post-menopausal female mice because most cancers are at old age. Post-menopausal female mice have high mortality per se, besides the treatment given by DOX causes moderate to severe suffering, which would radically increase death rate. The use of female mice raises another issue, if we used adult females instead: their hormonal cycle. If we used pre-menopausal mice induced estrous cycle synchronization for more reproducible results would be done, either by: 1) exposing females to male urine, so the majority go into estrous on the 3rd day (Whitten effect) or 2) injecting progesterone on day 1 and 2, synchronizing the estrous cycles of most females (about 85%) at metestrus on day 3 [6].  These changes would be a bias also, because the female population is not on this permanent state. In fact, most published studies use male mice [7-8], which appear to be as susceptible to anthracycline cardiotoxicity [9-11], and therefore, males were studied here as the aim of the study is to see the age impact on DOX cardiotoxicity.

COMMENTS OF REVIEWER 2: “second, the authors develop the procedure on some previously developed scale. My question is how many papers have used this scale? Is it commonly used for this type of experiments and if it is used, is it calculated on the basis of the average of several experimenters? my understanding is that all scales are inherently a somewhat subjective assessment”

AUTHOR’S RESPONSES TO REVIEWER 2:  Grimace scale is an excellent method to identify pain and is very helpful to maintain animal wellbeing in research studies [12]. The scores generated from the grimace scale should be used together with the context in which the animal is scored, its history, the procedure performed and the general parameters for wellbeing (sex, strain, species) [12]. In this work, it was adapted by authors to incorporate critical aspects and expected side effects of the treatment (DOX). The scale of pain was developed after other data was obtained and was approved by national officials (reference number 0421/000/000/2016) with a help of a Veterinary with a master’s in science and Welfare of Laboratory Animals (FELASA D category). Moreover, since it is a subjective assessment, the daily evaluation of animal’s welfare was performed by two different people with broad experience in animal behaviour.

COMMENTS OF REVIEWER 2: “Carbonylation of proteins as a measure of ROS is the first time, I've encountered such an assessment...would the authors consider confirming with another method. it doesn't appeal to me.... there is a slot dot measurement where by design of the method we have no loading counter”.

AUTHOR’S RESPONSES TO REVIEWER 2: We thank you for the reviewer’s comment. We clarified in the manuscript that protein carbonylation is a consequence of oxidative stress in the text.

Line 870-871 page 21:Protein carbonylation is a frequent type of ROS-induced protein modification, being considered irreversible and aims to induce protein degradation [54]”.

Regarding the slot dot measurement, 7 adult animals of each condition were selected. In the analysis, we used 14 samples plus a duplicate control. For statistical analysis we used 14 (7 from each condition).

We placed the image with the 14 samples on top of the graph and not the 12 as we had done before.

Changes in Line 496 page 13: Figure 5:

COMMENTS OF REVIEWER 2: “While we are on the subject of loading control, the same is true of Ponceau staining ... it is rather used classically to check whether the transfer was successful ... most often a measurement of the level of a reference protein e.g., actin is added and during densitometric analysis it is recalculated against it and this should be on the figure”.

AUTHOR’S RESPONSES TO REVIEWER 2: We thank the reviewer’s comment, and we acknowledge the reviewer´s point of view. Nevertheless, there is no uniformity regarding the best presentation for the Western blotting results and their normalization among the literature. We, as others [13-15], have chosen other normalization regarding cardiotoxicity of anticancer drugs [3, 16], since classical loading markers [e.g. tubulin, β-actin, glyceraldehyde-3-phosphate dehydrogenase (GAPDH) and so on] are changed due to cardiac muscle adaptations and further exacerbated in heart diseases involving cellular hypertrophy [17-21]. Moreover, in an in vivo work developed by our group, aged control animals exhibited increased levels of GAPDH [16] when compared to adult controls. Therefore, we have chosen Ponceau S staining for proof of equal loading and we add the data of Ponceau S staining for each Western blot performed in supplementary data as the reviewer can kindly see. The stainings are uniform between lanes and give similar loading.

COMMENTS OF REVIEWER 2: “I honestly don't understand what the authors are presenting on the y-axis in their analysis of western blotting??? not to mention the fact that in the figures it should be tested stripe and control of loading.... captions of which protein level we are testing should be in the title and not a cluster of graphs and it is on the y-axis or in the caption of the figure that I have to look for what was tested here, in addition, some photos have terrible pixelation.... I would like to see whole membranes. the quality of some photos is unacceptable, and I cannot even say which ones because figures 9 and 10 do not have a division into panels”

AUTHOR’S RESPONSES TO REVIEWER 2: We thank the reviewer’s comment, and we acknowledge the reviewer´s point of view. Regarding the loading, we have responded in the previous point. Regarding the captions of which protein used, we now placed it in title of graphic as suggested by the reviewer.

Changes in Line 576-577 page 17: Figure 9:

Changes in Line 633-634 page 18: Figure 10:

In addition, we formatted the image with "advanced metafile with the highest quality possible".

Moreover, we provide next an example the full membranes for the reviewers’ inspection.

COMMENTS OF REVIEWER 2: “discussion as if written backwards - pointing out what new has been discovered as a form of summary is ok but as a discussion starter it simply does not look like it”.

AUTHOR’S RESPONSES TO REVIEWER 2: To comply with the suggestion given, we changed the summary of discussion and we followed that order in the discussion section.

Line 642-659 page 18: In the present work, the early myocardial changes caused by DOX were assessed in mice of two ages and the major findings were: (1) DOX affected animals’ body weight and food/water consumption; (2) DOX caused a significant elevation of plasma levels of CK-MB, total-CK, AST and AST/ALT ratio in adult mice, which was corroborated by histopathological findings; (3) the ratio between heart weight/brain weight was significantly decreased in both populations; (4) the levels of macrophage M1 marker and of NF-κB p65 subunit increased in DOX-treated adult animals; (5) no differences were found in the levels of TNF-α, IL-1β, p38 MAPK, and NF-κB p52 in both evaluated populations; (6) in DOX-treated adult animals, the levels of IL-6 significantly decreased; (7) in DOX-treated adults an increase of fibrotic tissue was observed, probably due to the activation of NF-κB transcription factor, while this last was possibly triggered by TNFR2 activation; (8) different responses were seen regarding protein carbonylation after DOX administration in infants and adults; (9) in DOX-treated adult animals, the levels of MPO significantly decreased, perhaps because early damage to the enzyme may lead to less subsequent oxidative stress (namely protein carbonylation), simultaneously blunting the protective Nrf2 activation. On the other hand, in infant mice, increased protein carbonylation leads to activation of Nrf2, which appears to protect the heart, meaning less damage to the myocardium.”

COMMENTS OF REVIEWER 2: “The work is very similar in structure to the quoted item 17 written by the authors this year - a different compound was used... I have an impression that a whole series will be created or maybe already created. What is the authors' explanation for the submission to biomolecules? I am not sure it is right fit.”

AUTHOR’S RESPONSES TO REVIEWER 2: The dissertation of the first author of this paper is focused on cardio-oncology and the effect of age on heart after chemotherapy. Work 17 used mitoxantrone and now we used DOX, since these two molecules share similar clinical and toxicological output, but we are trying to highlight that they have different intracellular mechanisms. Age is the common link between the works, but different doses were used for the drugs under study. We are currently concluding our original research manuscript focused on DOX cardiotoxicity, which we believe would be suitable for the special issue "Cardiovascular Pharmacology" since DOX is used in several types of cancer and adverse side effects come with its use, being cardiotoxicity a limiting effect with serious and possible life-threatening consequences. It is often characterized by arrhythmias, electrocardiographic changes, and HF. The HF has a mortality rate of 50% in 5 years, with no planned curable approach [22]. In fact, on this subject (cardio-oncology) we have published before on Biomolecules two works:

  • The Main Metabolites of Fluorouracil + Adriamycin + Cyclophosphamide (FAC) Are Not Major Contributors to FAC Toxicity in H9c2 Cardiac Differentiated Cells. Reis-Mendes A, et al. Biomolecules. 2019. PMID: 30862114
  • Doxorubicin Is Key for the Cardiotoxicity of FAC (5-Fluorouracil + Adriamycin + Cyclophosphamide) Combination in Differentiated H9c2 Cells. Pereira-Oliveira M, Reis-Mendes A, et al. Biomolecules. 2019. PMID: 30634681

Moreover, we believe the manuscript is suitable for the special issue "Cardiovascular Pharmacology" were it was submitted, since according to the aim of this SI “ this Special Issue will accept original research articles and up-to-date reviews about all aspects of cardiovascular diseases, mainly focusing on heart failure, and that cover but are not restricted to molecules involved in…Disrupted intracellular pathways”…as our work aims.

Round 2

Reviewer 2 Report

Authors adressed all raised concerns.

Author Response

AUTHOR’S RESPONSES TO REVIEWER: We thank you for the reviewer’s comment.